# Exposure to Non-Antimicrobial Drugs and Risk of Infection with Antibiotic-Resistant Enterobacteriaceae

**DOI:** 10.3390/antibiotics12040789

**Published:** 2023-04-20

**Authors:** Meital Elbaz, Esther Stein, Eli Raykhshtat, Ahuva Weiss-Meilik, Regev Cohen, Ronen Ben-Ami

**Affiliations:** 1Department of Infectious Diseases, Tel Aviv Sourasky Medical Center, Tel Aviv 6423906, Israel; 2Faculty of Medicine, Tel Aviv University, Tel Aviv 6997801, Israel; 3I-Medata AI Center, Tel Aviv Sourasky Medical Center, Tel Aviv 6423906, Israel; 4Hillel Yaffe Medical Center, Hadera 3820302, Israel; 5Rappaport Faculty of Medicine, Technion, Haifa 3200003, Israel

**Keywords:** antimicrobial resistance, modeling, non-antimicrobial drugs

## Abstract

Antimicrobial resistance (AMR) has consistently been linked to antibiotic use. However, the roles of commonly prescribed non-antimicrobial drugs as drivers of AMR may be under-appreciated. Here, we studied a cohort of patients with community-acquired pyelonephritis and assessed the association of exposure to non-antimicrobial drugs at the time of hospital admission with infection with drug-resistant organisms (DRO). Associations identified on bivariate analyses were tested using a treatment effects estimator that models both outcome and treatment probability. Exposure to proton-pump inhibitors, beta-blockers, and antimetabolites was significantly associated with multiple resistance phenotypes. Clopidogrel, selective serotonin reuptake inhibitors, and anti-Xa agents were associated with single-drug resistance phenotypes. Antibiotic exposure and indwelling urinary catheters were covariates associated with AMR. Exposure to non-antimicrobial drugs significantly increased the probability of AMR in patients with no other risk factors for resistance. Non-antimicrobial drugs may affect the risk of infection with DRO through multiple mechanisms. If corroborated using additional datasets, these findings offer novel directions for predicting and mitigating AMR.

## 1. Introduction

Antimicrobial resistance (AMR) is a global health crisis that is estimated to result in 2.8 million infections and 35 thousand deaths in the US and at least 1.2 million deaths globally each year [1,2]. Among these, infections with drug-resistant Enterobacterales are associated with significant morbidity and mortality, and have been designated a serious threat by the US CDC [1]. Specifically, infections with extended-spectrum β-lactamase producing Enterobacterales (ESBL-PE) are associated with high overall and infection-related mortality, length of stay, and hospital costs, likely resulting from delayed initiation of appropriate antimicrobial therapy [3].

Acquired bacterial resistance to antimicrobials has been consistently linked to exposure to antibacterial drugs in humans and animals [4,5,6,7], making recent antibiotic exposure a core element of models for predicting antimicrobial resistance and decision-support tools for guiding empirical antibiotic treatment [6,8]. Nevertheless, this approach has limited clinical utility, as 30% to 50% of patients harboring drug-resistant Enterobacterales at hospital admission have no identifiable risk factors [6,7,8,9].

Many non-antimicrobial drugs (NAMD) have off-target antibacterial activities that affect the abundance and metabolism of multiple bacterial genera at clinically relevant concentrations [10]. In vitro susceptibility testing using anaerobic growth conditions showed significant inhibition of gut commensal bacteria by 24% of NAMDs [10]. Specific drug classes with significant antibacterial activity included proton pump inhibitors, atypical antipsychotics, calcium channel blockers, non-steroidal anti-inflammatory drugs, and antimetabolites [10,11]. Importantly, certain drug resistance mechanisms, such as efflux transporter expression, transcription factors regulating drug efflux and ribosomal maturation, and drug detoxification mechanisms, are shared between NAMDs and antimicrobial drugs, suggesting that exposure to non-antimicrobials may select for antimicrobial-resistant bacterial strains [10]. Moreover, beta-blockers, cytotoxic drugs, and non-steroidal anti-inflammatory drugs were found to promote bacterial transformation and facilitate the acquisition of antimicrobial resistance genes [11,12].

The role of exposure to NAMDs as a risk factor for infection with drug-resistant bacteria has not been systematically studied. Here, we aimed to explore the correlation between exposure to frequently prescribed NAMDs and infection with drug-resistant Enterobacterales in patients with acute community-onset pyelonephritis.

## 2. Results

We identified 3742 patients hospitalized with community-onset pyelonephritis within the study period. Out of these, 2310 had a positive urine or blood culture. We excluded 338 patients with urinary tract infections caused by non-fermenting Gram-negative organisms and Enterococcus spp., and 165 patients with hospital-acquired infections. Thus, the study cohort included 1807 patients (Table 1). Median patient age was 82 years (interquartile range, 71 to 88 years), 1065 (58.9%) were females, 358 (19.8%) had indwelling urinary catheters, and 520 (26.4%) had bloodstream infections.

Drug resistant organisms (DRO, Enterobacterales with at least one resistance phenotype) were identified in 944 (52.2%) of patient episodes: 611 (33.8%) resistant to ceftriaxone, 331 (18.3%) resistant to aminoglycosides, 692 (38.3%) resistant to ciprofloxacin, 612 (33.8%) resistant to trimethoprim-sulfamethoxazole (TMP-SMX), and 9 (0.5%) resistant to meropenem. Multidrug-resistant organisms (MDRO, ≥3 resistance phenotypes) were identified in 431 (23.8%) episodes. Patient covariates associated with resistance are shown in Table 2.

We studied the association between exposure to 19 NAMDs or classes and five antibiotic resistance phenotypes (Figure 1). On bivariate analysis, exposure to seven of the NAMDs was significantly associated with at least one resistance phenotype. NAMD and their respective associated antimicrobial resistance phenotypes were proton pump inhibitors (PPI; resistance to TMP-SMX, ciprofloxacin, ceftriaxone, and meropenem), antimetabolites (TMP-SMX, ciprofloxacin, ceftriaxone, and meropenem), beta-blockers (aminoglycosides, TMP-SMX, ciprofloxacin and ceftriaxone), clopidogrel (ceftriaxone), selective serotonin reuptake inhibitors (SSRI; ceftriaxone), typical anti-psychotics (aminoglycosides and meropenem), and anti-Xa anticoagulants (ceftriaxone and TMP-SMX). Exposure to PPI and beta-blockers was significantly associated with infection with MDRO.

The probability of exposure to each of the NAMDs conditional on patient age, comorbidities (Charlson score), functional status (Norton score), and previous hospitalization was computed using logistic regression (Table 3). Covariates that were significantly associated with NAMD exposure were used to calculate inverse probability weights. Treatment effects models were constructed with NAMD exposure as the treatment variable and antibiotic resistance as the outcome variable. Recent exposure to antimicrobials and urinary catheterization was associated with increased risk of all five resistance phenotypes (Table 3) and were entered as covariates in the final regression model.

Six of the seven NAMDs that increased the probability of antibiotic resistance in the bivariate analyses were significantly associated with DRO infection in the treatment effects model (Figure 2, Table 4). Beta-blockers, PPI, and antimetabolites were each associated with multiple (3 or 4) resistance phenotypes. SSRI and clopidogrel were associated with ceftriaxone resistance, and anti-Xa anticoagulants were associated with resistance to TMP-SMX. Five out of six NAMDs were associated with ceftriaxone resistance, four with TMP-SMX resistance, three with aminoglycoside resistance, and two with ciprofloxacin resistance. None of the NAMDs were associated with meropenem resistance, likely because of the rarity of this phenotype. Beta-blockers were the only NAMD significantly associated with MDRO infection.

The average treatment effects were 20.9 percentage points (range, 18.4 to 29.8 percent) for antimetabolites, 6.1 percentage points (range, 1.6 to 7.5 percent) for PPI, and 5.2 percentage points (range, 5.1 to 7.9 percent) for beta-blockers. Average treatment effects were 9.6%, 9.0%, and 9.0% for anti-Xa anticoagulants, clopidogrel, and SSRI, respectively (Figure 2, Table 4).

To gain further insight into the interaction between non-antimicrobials and patient characteristics, we analyzed the probability of resistance to ceftriaxone and TMP-SMX conditional on the status of the two patient covariates, urinary catheterization and recent treatment with antibiotic drugs (Figure 3). For all NAMDs that were significantly associated with ceftriaxone or TMP-SMX resistance, the probability of resistance increased for patients with negative urinary catheter and antibiotic exposure status (1333 patients, 73.8% of the cohort). The gain in resistance probability for this population ranged from 9 to 32 percentage points. Smaller increases in resistance probability were observed for patients with urinary catheters and no antibiotic exposure (327 patients, 18.1% of the cohort; 5 to 16 percentage points). The probability of resistance did not increase for patients with a history of antibiotic exposure (146 patients, 8% of the cohort), with two exceptions: SSRI and antimetabolites were associated with resistance to ceftriaxone and TMP-SMX, respectively, across all catheter and antibiotic exposure conditions (Figure 3).

## 3. Discussion

Predicting and mitigating infections with antibiotic-resistant organisms is an important goal of infection control and antibiotic stewardship programs. Antibiotics are powerful drivers of the emergence and dissemination of antibiotic-resistant bacterial strains; however, the role of non-antimicrobial agents may be overlooked. In this observational study of patients with community-acquired pyelonephritis, receipt of certain non-antimicrobial drugs was significantly associated with a higher probability of infection with antibiotic-resistant Enterobacterales. The weighted effect of exposure to non-antimicrobial drugs was of similar magnitude to that of exposure to antibiotics, and the effect was greatest for patients with no identifiable risk factors for DRO infection.

Out of 19 NAMDs assessed in this study, six (31%) were found to be independently associated with antibiotic resistance phenotypes. Two general patterns of association were observed. PPI, antimetabolites, and beta-blockers were associated with resistance to multiple antibiotics with different mechanisms of action. In contrast, clopidogrel, anti-Xa agents, and SSRIs were each associated with a single resistance phenotype. The three NAMD classes that were associated with multiple resistance phenotypes were previously linked to bacterial alterations that could potentially lead to the acquisition of antibiotic resistance. The beta-blocker propranolol was shown to enhance the horizontal transfer of antibiotic resistance genes among bacteria by facilitating transformation [11]. Propranolol increased the uptake of cell-free DNA fragments carrying antibiotic resistance genes by increasing bacterial cellular stress, membrane permeability, and competence. Antimetabolites are cytotoxic agents whose cellular targets are often conserved in bacteria [13]. These agents were found to inhibit gut commensals at clinically relevant concentrations [10], and were associated with reduced microbial diversity in the Flemish cohort study [14]. Moreover, cytotoxic drugs drive bacterial mutagenesis by inflicting DNA damage and activating the bacterial SOS response [12]. DNA repair occurring as part of this response is mediated by low-fidelity polymerases and is associated with de novo mutations [15,16]. Cleavage of the SOS repressor LexA and de-repression of polymerases were shown to be essential to the development of fluoroquinolone resistance in *E. coli* [17]. The SOS response also enhances horizontal gene transfer, facilitating the simultaneous acquisition of multiple antimicrobial resistance genes on mobile genetic elements [12]. PPI was also shown to directly inhibit drug commensals [10] and to alter the gut microbiome [18,19]. In addition, PPI was found to induce resistance to the antimicrobial tigecycline in a concentration-dependent manner [20]. A meta-analysis of 26 studies including more than 29,000 participants showed that the use of these agents was associated with a ~75% increase in the odds of MDRO colonization [21]. The risk was higher with PPI than with H2-receptor antagonists, suggesting that the degree of acid suppression is a driver of bacterial resistance. Interestingly, the ESBL-producing *E. coli* sequence type 131 is relatively resistant to gastric acid [22], and might preferentially colonize the lower intestinal tracts of patients with PPI-induced hypochlorhydria.

Intestinal carriage of drug-resistant Gram-negative bacteria is on the rise among non-hospitalized populations. A systematic review and meta-analysis showed a cumulative prevalence of 16.5% carriage of ESBL-producing *E. coli* [23]. The prevalence of carriage trended upward from 2.6% in 2003–2005 to 21.1% in 2015–2018. Studies of risk factors for antimicrobial resistance typically do not assess exposure to NAMD, with the exception of PPI. However, the identification of certain chronic diseases as risk factors for antimicrobial resistance may serve as a proxy for NAMD exposure. A recent survey conducted on non-hospitalized individuals in sub-Saharan Africa found almost universal carriage of ESBL-PE, irrespective of antibiotic consumption [24]. The authors hypothesized that the high prevalence of chronic diseases, such as hypertension and diabetes mellitus, as well as transmission of ESBL-PE among household members and livestock, might explain this finding. Similarly, infection with fluoroquinolone-resistant *E. coli* was found to be linked to diabetes mellitus, cardiovascular disease, and heart failure, all of which might be associated with exposure to drugs identified in our study as risk factors for antimicrobial resistance, such as clopidogrel, beta-blockers, and anti-Xa inhibitors [25].

The NAMDs associated with DRO infections in our study partially overlap with drug classes identified as having in vitro activities against gut commensals [10]. However, some drug classes with anti-commensal activities, such as calcium channel blockers and metformin, showed no association. Several explanations could be proposed to explain this discrepancy. First, these differences underscore the fact that in vitro drug-bacteria interactions do not necessarily predict the induction of antimicrobial resistance within a complex in vivo system consisting of bidirectional interactions between multiple bacterial communities within the gut microbiome, antimicrobial and non-antimicrobial drugs, and antibiotic resistance genes [26,27]. Second, NAMDs are generally handled by bacterial efflux transporters, which may share antibiotics and NAMDs as substrates [10]. However, efflux mechanisms do not play a major role in the resistance of Enterobacterales to beta-lactams, fluoroquinolones, or sulfonamides, the major resistance phenotypes analyzed in the present study. Third, the present analysis was limited to Enterobacterales, and therefore previously described activities of NAMD such as dabigatran and ticagrelor against Gram-positive bacteria [28,29] could not be captured. Lastly, in contrast to studies on microbiome composition, the primary outcome of the current study was upper urinary tract infections with drug-resistant Enterobacterales, an endpoint that could reflect additional interactions between NAMD use and bacterial virulence and host susceptibility to infection.

Some limitations of the present study should be noted. The study’s single-center design might limit the generalizability of the findings. Moreover, dose response could not be assessed, because the dataset did not include information on the dose and duration of non-antimicrobial drugs. Finally, we did not examine the effects of drug combinations. Thus, our analysis might not detect synergistic interactions between NAMD and antibiotic drugs [30].

In summary, we found significant interactions between exposure to non-antimicrobial drugs and multiple resistance phenotypes in patients with community-acquired pyelonephritis. These findings should be validated using additional patient datasets from diverse settings. Considering the clinical and epidemiological importance of infections with drug-resistant Enterobacterales, incorporating specific non-antimicrobials into schemes to predict and mitigate AMR could have far-reaching implications.

## 4. Methods

### 4.1. Study Overview

This was an observational cohort study of patients admitted into hospital with community-onset pyelonephritis. The study was performed at the Tel Aviv Sourasky Medical Center (TASMC), a 1500-bed tertiary-level academic hospital in Tel Aviv, Israel.

Patients meeting study inclusion criteria were identified by querying the electronic medical record system and computerized microbiology laboratory databases. We included hospitalized adult (≥18 years) patients who were discharged from internal medicine and geriatric departments between 1st January 2017 and 18th April 2019, with a diagnosis of upper urinary tract infection (ICD9 codes: 599.0, 590.1, 590.8, 590.80, and 590.9). Patients were included if they had urine or blood culture growing *E. coli*, *Klebsiella* spp., *Proteus* spp., *Enterobacter* spp., *Citrobacter* spp., *Serratia* spp., *Providencia* spp., or *Morganella* spp. Only patients with community-onset infection, defined as index culture specimens obtained <48 h after hospital admission, were included.

Patients were excluded if they had mixed bacterial growth on their urine culture or had a urine or blood culture growing bacteria other than Enterobacterales (e.g., *Enterococcus* spp or non-fermenting Gram-negative organisms, such as *Pseudomonas* spp. and *Acinetobacter* spp.). Patients with a hospital-acquired urinary tract infection, defined as index culture collected >48 h after admission, were excluded.

The primary outcome was infection with an antimicrobial drug-resistant organism. Independent variables were prior exposure to NAMD, and confounders were patient variables associated with either likelihood of NAMD exposure or infection with DRO. Treatment effects models were constructed to account for potential biases associated with NAMD exposure.

The study was reviewed and approved by the TASMC ethics committee (approval number 0821-18-TLV). All procedures were performed in accordance with relevant guidelines and regulations. Requirement for informed consent was waived considering the retrospective observational nature of the study.

### 4.2. Non-Antimicrobial Drug Exposure

Exposure to NAMD prior to hospital admission was retrieved from electronic medical record documentation. NAMD were categorized by class and activity. The following NAMD classes were assessed: angiotensin-converting enzyme inhibitors; angiotensin-receptor blockers; dihydropyridine and non-dihydropyridine calcium channel blockers; beta-blockers; aspirin; clopidogrel; statins; direct thrombin inhibitors including dabigatran; anti-factor Xa agents including apixaban and rivaroxaban; metformin; sulfonylurea; proton pump inhibitors; H2 receptor-blockers; anti-metabolites (folic acid analogs, purine analogs, and pyrimidine analogs); typical and atypical anti-psychotic agents; tricyclic anti-depressants; selective serotonin reuptake inhibitors; and serotonin–norepinephrine reuptake inhibitors (SNRI).

Additional covariates included recent hospitalization during the previous 100 days prior to collection of index culture, indwelling urinary catheter, Norton functional score, Charlson comorbidity score [31], and previous exposure to antibiotic treatment.

### 4.3. Microbiological Methods

Semi-quantitative urine cultures were performed using the Diaslide device (Novamed, Jerusalem, Israel), according to the manufacturer’s instructions. Urine was streaked onto non-selective enriched chromogenic medium (URIselect) and MacConkey agar and incubated at 35 °C for 18 to 24 h. Colony densities consistent with ≥10^3^ bacteria/mL were reported as positive. Blood culture bottles were incubated in the BacT/Alert Virtuo system (bioMerieux, Marcy L’Etoile, France) for up to 5 days. Bacteria were identified using the Vitek 2 system (bioMerieux), and antibacterial susceptibility testing was performed using Vitek 2 and Clinical and Laboratory Standards Institute breakpoints [32].

DROs were classified according to 5 resistance phenotypes: aminoglycoside (gentamicin or amikacin), ciprofloxacin, trimethoprim-sulfamethoxazole (TMP-SMX), ceftriaxone, and meropenem. Multidrug-resistant organisms were defined as having at least 3 of the resistance phenotypes [33].

### 4.4. Statistical Analyses

Frequency of exposure to each NAMD entity was compared among patients with and without each antibiotic resistance phenotype using Fisher’s exact test. NAMDs that were found to be significantly linked to antibiotic resistance were further assessed using a treatment effects estimator that models both outcome and treatment probability.

The treatment model was constructed in 3 steps [34]. First, the probability of exposure to each NAMD conditional on patient covariates was calculated using binomial logistic regression. Covariates that were significantly associated with NAMD exposure were used to calculate inverse probability weights. Second, separate binary logistic regression models of the probability of resistance were calculated for each NAMD exposure level for each subject. Finally, the weighted means of the exposure-specific predicted probability of resistance was computed, where the weights are the inverse-probability weights computed in the first step. The difference between these weighted averages provided estimates of the average NAMD treatment effect. The overlap assumption, which states that each individual has a positive probability of receiving treatment or control, was checked by plotting treatment probability densities for each model.

A type I error of <0.05 was considered statistically significant. Calculations were performed in Stata 15.0 (Statacorp, College Station, Texas). Graphs were plotted with GraphPad Prism version 6.0 (GraphPad Software, La Jolla, CA, USA).

## Figures and Tables

**Figure 1 antibiotics-12-00789-f001:**
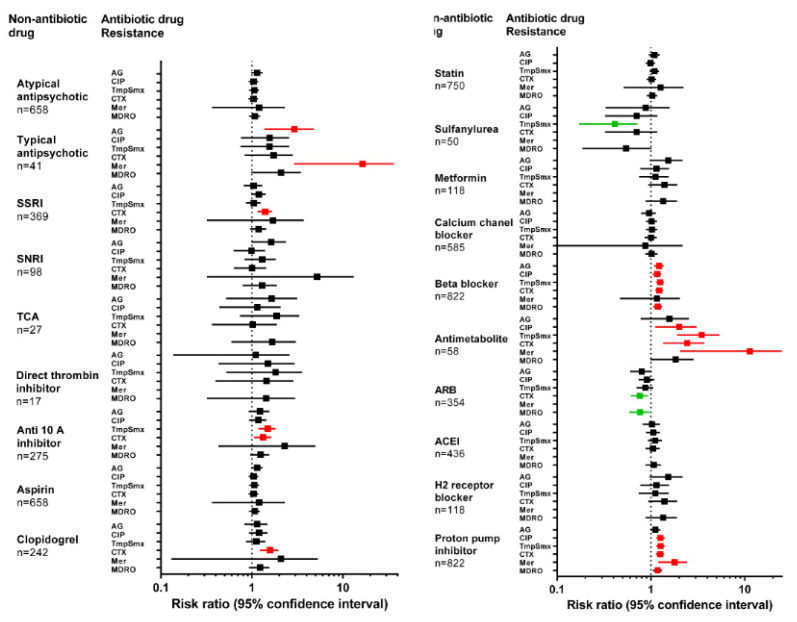
Risk of antimicrobial resistance and non-antimicrobial drug exposure. The forest plot shows the risk of infection with a drug-resistant organism expressing each of 5 resistance phenotypes, as a function of recent exposure to non-antimicrobials. Squares represent odds ratio, and whiskers represent the 95% confidence interval. Markers in red and green represent significantly increased or decreased risk of resistance phenotype, respectively. AG: aminoglycoside; CIP: ciprofloxacin; Tmp-Smx: trimethoprim-sulfamethoxazole; CTX: ceftriaxone; Mer: meropenem; MDRO: multidrug resistant organism; SSRI: selective serotonin reuptake inhibitor; SNRI: serotonin and norepinephrine reuptake inhibitor; TCA: tricyclic antidepressant; ARB: angiotensin receptor blocker; ACEI: angiotensin converting enzyme inhibitor.

**Figure 2 antibiotics-12-00789-f002:**
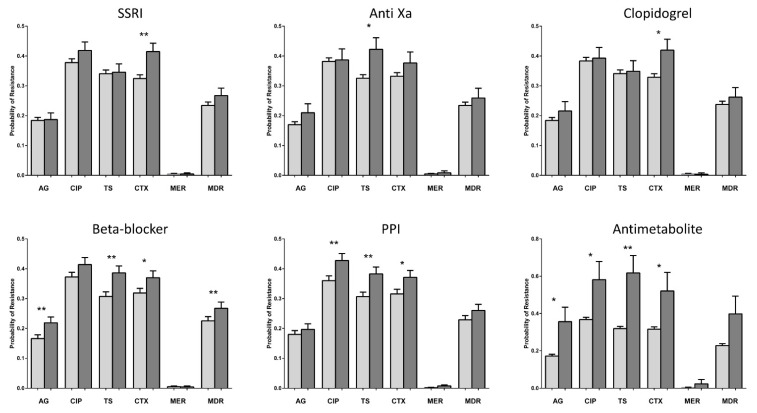
Treatment effects model of antimicrobial resistance and non-antimicrobial drug exposure. Bars represent odds of infection with an organism expressing the resistance phenotype calculated from the treatment effects model (see Methods). Error bars represent the 95% confidence interval. SSRI: selective serotonin reuptake inhibitor; PPI: proton pump inhibitor; AG: aminoglycoside; CIP: ciprofloxacin; TS: trimethoprim-sulfamethoxazole; CTX: ceftriaxone; MER: meropenem; MDR: multidrug resistant organism. * *p* < 0.05; ** *p* < 0.01.

**Figure 3 antibiotics-12-00789-f003:**
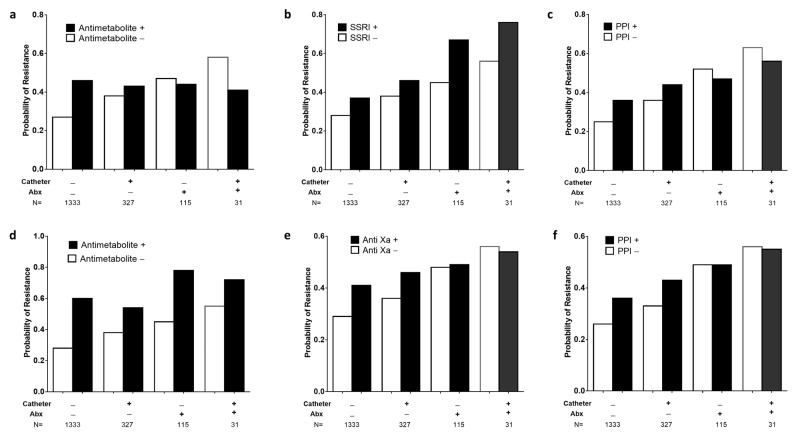
Effect of patient covariates and non-antimicrobial treatment on probability of antibiotic resistance. Bars show the treatment effects model derived probability of antibiotic resistance phenotype, as a function of exposure to non-antimicrobial drugs and 2 patient covariates, indwelling urinary catheter (Catheter) and exposure to antibacterial agents (Abx). Panels (**a**–**c**) show probability of resistance to ceftriaxone, and panels (**d**–**f**) show resistance to trimethoprim-sulfamethoxazole. SSRI: selective serotonin reuptake inhibitor; PPI: proton pump inhibitor.

**Table 1 antibiotics-12-00789-t001:** Characteristics of 1806 patients with community-acquired pyelonephritis.

Characteristic	All Patients
Age, years, years, median (IQR)	82 (71–88)
Sex	
Male	742 (41.1)
Female	1065 (58.9)
Charlson comorbidity score, median (IQR)	5 (4–7)
Norton score, median (IQR)	16 (13–20)
eGFR, ml/min, median (IQR)	51.7 (33.2–75.2)
Bloodstream infection	520 (26.4)
Inotropic support	65 (3.3)
Indwelling urinary catheter	358 (19.8)
Microbiology	
*E. coli*	1239 (62.83)
*Klebsiella* spp.	417 (21.15)
*Proteus* spp.	177 (8.98)
Other Enterobacterales	139 (7.05)
DRO	
Aminoglycoside	331 (18.3)
Ciprofloxacin	692 (38.3)
TMP-SMX	612 (33.87)
Ceftriaxone	611 (33.81)
Meropenem	9 (0.5)
MDRO	431 (23.8)

Categorical variables are presented as number of patients (percent) and continuous variables are presented as median (interquartile range). eGFR, estimated glomerular filtration rate; DRO, drug-resistant organism; MDRO: multidrug resistant organism; IQR, interquartile range.

**Table 2 antibiotics-12-00789-t002:** Association between patient covariates and antibiotic resistance phenotypes.

Resistance Phenotype	AG *n* = 331	Ceftriaxone *n* = 611	Ciprofloxacin *n* = 692	Meropenem *n* = 9	Tmp-Smx *n* = 612	MDRO *n* = 431
Age	1.00 (0.99–1.01)	**1.01** **(1.00–1.01)**	**1.01** **(1.00–1.02)**	0.99 (0.95–1.02)	1.00 (0.99–1.00)	**1.00** **(0.99–1.01)**
Norton score	**0.91** **(0.88–0.94)**	**0.88** **(0.86–0.9)**	**0.9** **(0.88–0.92)**	0.87 [0.72–1.04]	**0.94** **(0.91–0.96)**	**0.90** **(0.87–0.92)**
CCI	1.03 (0.98–1.08)	**1.13** **(1.09–1.18)**	1.12 (1.081–1.16)	1.04 (0.81–1.35)	**1.09** **(1.04–1.13)**	**1.08** **(1.04–1.13)**
Recent hospitalization	**1.59** **(1.39–1.82)**	**1.6** **(1.43–1.84)**	**1.4** **(1.24–1.6)**	**2.3** **(1.61–3.28)**	**1.39** **(1.22–1.58)**	**1.59** **(1.4–1.8)**
Previous antibiotic therapy	**2.25** **(1.63–3.1)**	**2.12** **(1.55–2.89)**	**1.79** **(1.31–2.45)**	1.37 (0.21–8.79)	**2** **(1.47–2.73)**	**2.2** **(1.63–3.03)**
Indwelling urinary catheter	**1.79** **(1.48–2.1)**	**1.4** **(1.18–1.71)**	**1.48** **(1.23–1.79)**	0.55 (0.087–3.55)	**1.31** **(1.08–1.58)**	**1.54** **(1.22–0.46)**

AG, aminoglycosides (gentamicin or amikacin); CCI: Charlson comorbidity index. Data shown as odds ratio (95% confidence interval). Bold for results with *p* value < 0.05.

**Table 3 antibiotics-12-00789-t003:** Patient variables associated with non-antimicrobial drug exposure.

Drug/Class	Age	Charlson Index	Norton	Hospitalization
Typical antipsychotic	**0.95** **(0.93–0.97)**	1.060 (0.89–1.25)	**0.82** **(0.76–0.89)**	1.15 (0.56–2.36)
SSRI	**1.012** **(1.00051–1.024)**	**1.071** **(1.0060–1.14)**	0.95 (0.92–0.98)	1.19 (0.91–1.57)
Anti Xa agents	**1.031** **(1.015–1.047)**	**1.16** **(1.08–1.25)**	0.99 (0.95–1.02)	1.29 (0.95–1.76)
Clopidogrel	0.99 (0.98–1.011)	**1.25** **(1.16–1.35)**	0.96 (0.92–1.00)	1.24 (0.89–1.74)
Beta-blocker	**1.016** **(1.0069–1.026)**	**1.14** **(1.085–1.20)**	1.0048 (0.97–1.031)	**1.34** **(1.062–1.69)**
Antimetabolite	**0.95** **(0.93–0.97)**	**1.20** **(1.046–1.38)**	**1.11** **(1.027–1.21)**	**3.38** **(1.046–1.38)**
Proton pump inhibitor	1.0089 (0.99–1.018)	**1.19** **(1.13–1.26)**	1.0058 (0.98–1.032)	**1.35** **(1.07–1.70)**

SSRI, selective serotonin reuptake inhibitor. Data shown as odds ratio (95% confidence interval). Bold for results with *p* value < 0.05.

**Table 4 antibiotics-12-00789-t004:** Treatment effects models.

Drug/Class	Resistance Phenotype	AG *n* = 331	Ceftriaxone *n* = 611	Ciprofloxacin *n* = 692	Meropenem *n* = 9	Tmp-Smx *n* = 612
SSRI	Probability unexposed	0.18	0.32	0.37	0.0048	0.34
	Average treatment effect	0.0028 ±0.022	0.090 ±0.028	0.040 ±0.028	−0.0003 ±0.0037	0.0052 ±0.027
	*P* value	0.89	0.001	0.15	0.93	0.85
Anti Xa agents	Probability unexposed	0.17	0.33	0.38	0.0044	0.32
	Average treatment effect	0.030 ±0.03	0.044 ±0.037	0.0053 ±0.037	0.0040 ±0.0064	0.096 ±0.038
	*P* value	0.3	0.23	0.88	0.52	0.012
Clopidogrel	Probability unexposed	0.18	0.32	0.38	0.0051	0.34
	Average treatment effect	0.031 ±0.031	0.090 ±0.036	0.0096 ±0.036	−0.0012 ±0.0042	0.0076 ±0.035
	*P* value	0.31	0.013	0.79	0.77	0.83
Beta-blockers	Probability unexposed	0.16	0.31	0.37	0.0053	0.30
	Average treatment effect	0.052 ±0.019	0.051 ±0.022	0.041 ±0.023	−0.00083 ±0.0033	0.079 ±0.023
	*P* value	0.006	0.026	0.079	0.80	0.001
Proton pump inhibitor	Probability unexposed	0.17	0.31	0.36	0.0019	0.30
	Average treatment effect	0.016 ±0.018	0.055 ±0.022	0.067 ±0.023	0.0060 ±0.0033	0.075 ±0.023
	*P* value	0.37	0.015	0.004	0.07	0.001
Antimetabolites	Probability unexposed	0.17	0.31	0.36	0.0039	0.31
	Average treatment effect	0.18 ±0.077	0.20 ±0.099	0.21 ±0.097	0.019 ±0.023	0.29 ±0.093
	*P* value	0.017	0.04	0.028	0.39	0.001

AG: aminoglycoside; SSRI, selective serotonin reuptake inhibitor. Treatment effect shown as change in probability from unexposed ± standard error of means.

## Data Availability

The datasets generated during the current study are available from the corresponding author on reasonable request.

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
