# Peer review of "Exposure to Non-Antimicrobial Drugs and Risk of Infection with Antibiotic-Resistant Enterobacteriaceae"

_antibiotics, 2023, doi:10.3390/antibiotics12040789_

Round 1

Reviewer 1 Report

Meital Elbaz and colleagues investigated the effects of non-antibiotic compounds on infections associated with antimicrobial resistance. This study has interest for understanding the possible co-selection effect in the spread of AMR, particularly critical pathogens of Enterobacteriaceae. While this is a pilot study, it could pave the way for future clinical studies that assess the mechanisms. Recently, pharmaceutics (non-antibiotics) have been noted to exert co-selection pressure on bacterial communities, not just antibiotics, which provide direct selection.  No major comments, but I wish the introduction had included more information about Enterobacteriaceae. In addition, if they have information on molecular analysis of plasmids and resistance genes, they may consider it in future studies.

Author Response

Meital Elbaz and colleagues investigated the effects of non-antibiotic compounds on infections associated with antimicrobial resistance. This study has interest for understanding the possible co-selection effect in the spread of AMR, particularly critical pathogens of Enterobacteriaceae. While this is a pilot study, it could pave the way for future clinical studies that assess the mechanisms. Recently, pharmaceutics (non-antibiotics) have been noted to exert co-selection pressure on bacterial communities, not just antibiotics, which provide direct selection. 

We thank the reviewer for their useful comments.

No major comments, but I wish the introduction had included more information about Enterobacteriaceae. In addition, if they have information on molecular analysis of plasmids and resistance genes, they may consider it in future studies.

As requested by the reviewer, we added a section summarizing current trends and risk factors for carriage of drug-resistant Enterobacterales in the community (lines 203-217 ).

In addition, the revised manuscript includes a more in-depth review of the effects of cytotoxic drugs on bacterial mutagenesis and horizontal gene transfer. The role of the drug-induced bacterial SOS response in derepression of error-prone DNA polymerases and plasmid transfer is discussed (lines 185-192).

Reviewer 2 Report

Meital Elbaz and colleagues explores the association between exposure to non-antimicrobial drugs and the risk of infection with antibiotic-resistant Enterobacteriaceae. The study suggests that commonly prescribed non-antimicrobial drugs may have important effects on the emergence and dissemination of antibiotic resistance of Enterobacterales in patients with infections. Therefore, the findings highlight the need for further research to explore the potential role of non-antimicrobial drugs as drivers of antimicrobial resistance and to develop strategies to mitigate this risk.

Put the results section before the methods section, the article is clear and original. But the discussion is not deep enough and needs to be deepened.

Line 112: Please insert relevant references.

Author Response

Meital Elbaz and colleagues explores the association between exposure to non-antimicrobial drugs and the risk of infection with antibiotic-resistant Enterobacteriaceae. The study suggests that commonly prescribed non-antimicrobial drugs may have important effects on the emergence and dissemination of antibiotic resistance of Enterobacterales in patients with infections. Therefore, the findings highlight the need for further research to explore the potential role of non-antimicrobial drugs as drivers of antimicrobial resistance and to develop strategies to mitigate this risk.

We thank the reviewer for their useful comments.

Put the results section before the methods section.

The order of sections within the manuscript was changed as per the reviewer’s instructions.

The article is clear and original. But the discussion is not deep enough and needs to be deepened.

The discussion has been expanded to include a more in-depth review of the effects of cytotoxic drugs on bacterial mutagenesis and horizontal gene transfer (lines 185-192), and a new section summarizing current trends and risk factors for carriage of drug-resistant Enterobacterales in the community (lines 203-217). We also address the possibility that known interactions between certain chronic diseases, such as cardiovascular disease, diabetes mellitus and heart failure, and antimicrobial resistance, could be partially explained by exposure to human-targeted drugs used for those conditions.

Line 112: Please insert relevant references.

As suggested, the reference for CLSI breakpoints was added to the Methods section (Line 300 in the revised manuscript).

Round 2

Reviewer 2 Report

No comments